# Convergent Genomic Signatures of High-Altitude Adaptation among Six Independently Evolved Mammals

**DOI:** 10.3390/ani12243572

**Published:** 2022-12-16

**Authors:** Tianshu Lyu, Shengyang Zhou, Jiaohui Fang, Lidong Wang, Lupeng Shi, Yuehuan Dong, Honghai Zhang

**Affiliations:** 1College of Wildlife and Protected Area, Northeast Forestry University, Harbin 150000, China; 2College of Life Science, Qufu Normal University, Qufu 273165, China

**Keywords:** comparative genomic, high-altitude, mammals, adaption, convergent evolution

## Abstract

**Simple Summary:**

In this study, we conducted a comparative genomics approach to explore the convergent evolution of high-altitude adaptation mechanisms in six independently evolved mammals belonging to four orders. The results showed that the convergent evolution of the six species was mainly reflected at the level of rapidly evolving genes, and the functions of these rapidly evolving genes were mainly related to hypoxia response and DNA damage repair. In addition, we found that the convergence of the gene family contraction of high-altitude species is much greater than that of expansion, revealing a possible pattern of species in adapting to high-altitude. Furthermore, we detected a positive selection signature in four genes related to hypoxia response and ultraviolet radiation damage in these six species (*FYCO1*, *ERBIN*, *SCAMP1* and *CXCL10*). In general, our study reveals that hypoxia response and UV-radiation might play an important role in the adaptation of independently evolved species to a high-altitude environment, providing a basic perspective for further exploring the high-altitude adaptation mechanism of different related species in the future.

**Abstract:**

The species living in the Qinghai-Tibet Plateau provide an excellent model system for studying the relationship between molecular convergent evolution and adaptation. Distant species experiencing the same selection pressure (i.e., hypoxia, low temperature and strong ultraviolet radiation) are likely to evolve similar genetic adaptations independently. Here, we performed comparative genomics studies on six independently evolved high-altitude species. The results also showed that the convergent evolution of the six species was mainly reflected at the level of rapidly evolving genes, and the functions of these rapidly evolving genes were mainly related to hypoxia response and DNA damage repair. In addition, we found that high-altitude species had more gene family changes than their low-altitude relatives, except for the order Lagomorpha. The results also show that the convergence of the gene family contraction of high-altitude species is much greater than that of expansion, revealing a possible pattern of species in adapting to high-altitude. Furthermore, we detected a positive selection signature in four genes related to hypoxia response and ultraviolet radiation damage in these six species (*FYCO1*, *ERBIN*, *SCAMP1* and *CXCL10*). Our study reveals that hypoxia response might play an important role in the adaptation of independently evolved species to a high-altitude environment, providing a basic perspective for further exploring the high-altitude adaptation mechanism of different related species in the future.

## 1. Introduction

As the highest plateau in the world, the average altitude of the Qinghai-Tibet Plateau exceeds 4000 m. The harsh living environment of the Qinghai-Tibet Plateau exerts strong selection pressure (i.e., hypoxia, low temperature and strong ultraviolet radiation) on native species, which has driven striking phenotypic and genetic adaptations [1]. A large number of high-altitude species has been explored since whole genome sequencing was used to explore the high-altitude adaptation mechanism of Tibetans, which greatly enriched our understanding of the impact of a high-altitude environment on the phenotype and genetic changes of species [2]. For example, the genes showing a positive selection signature of the Tibetan antelope were mainly associated with energy metabolism and oxygen transmission [3]. The study on yak, another species belonging to the family Bovidae, showed that, in addition to the positive selection signal detected in the genes related to hypoxia stress, the genes related to nutrition metabolism also changed significantly [4]. In addition, a considerable number of studies have confirmed that hypoxia adaptation is key for species to adapt to the plateau environment, and independently evolved species might have different genetic alterations and mechanistic preferences in response to the same selection pressures [5,6,7,8].

Convergent evolution refers to the evolution of different lineages that are similar in structure or function, and this evolution has nothing to do with the existence of a common ancestor between species. Simply put, the fact that structure or function are similar does not reflect homology [9]. Strictly speaking, convergence at the molecular level refers to the substitution of the same alleles with independent origins in different lineages—that is to say, convergent evolution refers to the independent conversion of different ancestral amino acids to the same derived amino acids at specific sites [10]. In broad terms, however, convergence may exist at the phenotypic or functional level, in which lineages share dominant traits, but the molecular mechanisms that achieve this convergence trait are different between species [11]. Convergent evolution occurs when species occupy similar ecological niches and adapt in similar ways to similar selection pressures [12,13]. For example, the research on giant panda and red panda found that the common specialized bamboo diet led to the adaptive convergence of two genes (*DYNC2H1* and *PCNT*) related to limb development [14]. Not only could the same diet habits lead to the convergent evolution of different species, but also the same living habits could lead to the eco-morphological convergence of different species. Previous studies found that there was eco-morphological convergence in slow arboreal xenarthrans through the analysis of multiple omics of the humeral and femoral internal structure [15]. Independently evolving species might lead to the convergent evolution of phenotypes in order to adapt to the same environment, but these convergent phenotypes do not necessarily result from convergent evolution at the molecular level. A comparative genomics study of three aquatic mammals (the killer whale, walrus and manatee) showed that although convergent phenotypic evolution can result from convergent molecular evolution, this is rare, and evolution more frequently utilizes different molecular pathways to achieve the same phenotypic outcome [16]. Convergent evolution is not only reflected in the phenotypic convergence of species but might also be reflected in the functional convergence at the molecular level. Previous analysis of the *CDH23* gene and its ligand *PCDH15* gene, as well as *OTOF* gene of neural signaling, found that these three genes had significantly convergent evolution in echolocation species (small bats and some whales), and *CDH23* and *PCDH15* genes had positive selection in echolocation species [17].

To date, a large number of studies have confirmed that the selection pressure exerted by the Qinghai-Tibet Plateau on indigenous species will also lead to the genetic changes of convergence at the molecular level. The genomes of Tibetan antelope showed that 247 positive selection genes related to hypoxia in Tibetan antelope and American pika had convergent evolution [3]. Genomic analysis also revealed a large number of convergent amino acid site replacements in four high-elevation anuran species from the Tibetan region [18]. The population genetic analysis showed that *EPAS1* and *HBB* genes related to hypoxia adaptation have undergone significant convergent evolution in Tibetans and Tibetan Mastiffs [19]. As the most well-known gene related to hypoxia response, the *EPAS1* gene also showed positive selection signals of convergent evolution in six domestic animals living on the plateau [11]. This study also reveals a fact—that is, the more species selected and the larger the genetic relationship span, the less evidence of convergent evolution at the molecular and genetic levels. Another study also confirmed that the parallel substitution of only one gene in four high-altitude animals resulted in a convergent heart phenotype [20]. Comparative transcriptomics is also applied to explore the convergent adaptation mechanism of species to high altitude. A previous comparative transcriptomic study of five high-altitude domestic animals and low-altitude related species showed that the differentially expressed genes at different altitudes might be related to the phenotypic differences caused by high-altitude adaptation [21]. Although the convergent adaptation mechanism of a large number of high-altitude species has been explored, the coverage is still relatively limited, and most of them focus on species with a close relationship (same genus or family) or domesticated species. It is not clear whether the mammalian species that have evolved independently and adapted to the plateau environment at different times could produce convergent molecular level or functional level changes when coping with the same selection pressure (i.e., hypoxia, low temperature and strong ultraviolet radiation) brought by high altitude.

In this study, we selected six representative mammals at the level of four orders, including Lagomorpha, Primates, Carnivora and Artiodactyla, in order to compare these six mammals with their low-altitude relatives through the method of comparative genomics, and explored their convergent evolution at the molecular or functional level when facing the selection pressure of high altitude. Among the six mammals, only the Tibetan sheep has artificially changed its living environment to adapt to the plateau, the yak was domesticated from the plateau wild species, while the rest of the wild species are naturally adapted to the high-altitude environment [3,4,5,22,23,24]. Furthermore, these six animals have great differences in size, feeding habits, phylogenetic relationships and life history. On the one hand, previous studies have shown that in order to cope with the intense selection pressures brought about by the Tibetan Plateau (such as low oxygen content, extreme temperature changes and high ultraviolet radiation), domesticated species and wild plateau species from different phylogenetic backgrounds have unique or identical high-altitude adaptation mechanisms [1,2,3,4,5,6,7,8,18,19]. On the other hand, the degree of phylogenetic relationships and the number of species selected also play important roles in the study of the convergence mechanism of species [11,20]. In other words, the more distantly related the species, the more likely it is that the convergent phenotype or function is the result of different molecular mechanisms, rather than the result of the same gene or amino acid substitution. Therefore, this study provides a new perspective on the mechanisms of high-altitude adaptation by exploring different mammals in the context of large-scale phylogeny and also expands the understanding of the impact of high altitude on the genetic changes in mammals.

## 2. Materials and Methods

### 2.1. Identification of Orthologous Gene Set

In order to study the characteristics of high-altitude mammals to the maximum extent, the genomes of six high-altitude species (*Rhinopithecus bieti*, *Pantholops hodgsonii*, *Ovis aries* (high-altitude), *Vulpes ferrilata*, *Ochotona curzoniae* and *Bos mutus*) and nine closely related low-altitude species (*Canis lupus familiaris*, *Bos taurus*, *Ovis aries* (low-altitude), *Capra hircus*, *Oryctolagus cuniculus*, *Bubalus bubalis*, *Trachypithecus francoisi*, *Vulpes lagopus* and *Piliocolobus tephrosceles*) were downloaded from the NCBI database (https://www.ncbi.nlm.nih.gov/, accessed on 5 March 2022) for subsequent analysis (Appendix A). OrthoFinder v2.4.0 (default) was used to identify orthogroups and orthologs among these 15 species [25]. Briefly, protein sets were collected from downloaded genomes of 15 species, and the longest transcripts of each gene were extracted, in which miscoded genes and genes exhibiting premature termination were discarded. The extracted protein sequences were then aligned pair-wise to identify conserved orthologs using Blastp set to an e-value threshold of ≤1 × 10^−5^, and orthologous inter-genome gene pairs, paralogous intra-genome gene pairs and single-copy gene pairs were further identified. Proteins with no homologs were extracted as species-specific genes including unique genes and unclustered genes.

### 2.2. Phylogenetic Analysis and Divergence Time Estimation

On the basis of the identified orthologous gene sets with OrthoFinder v2.4.0, molecular phylogenetic analysis was performed using the shared single-copy genes. Briefly, the coding sequences were extracted from the single-copy families, and each ortholog group was multiple aligned using Muscle v3.8.31 [26]. Then, we combined all the alignment results to form a super alignment matrix. Finally, the phylogenetic relationship of 15 species was constructed using the maximum likelihood method (ML Tree) based on the sequence alignment results using RAxML v8.2.12 (-m PROTGAMMAAUTO -p 12345 -T 8 -f b) [27].

The estimation of divergence time was performed using MCMCTree package in PAML v4.8 [28]. The generated tree file was displayed using FigTree v1.4.4 and MEGA v10.1.8 [29]. The silhouettes of the species with no copyright were download from the professional silhouette website (http://phylopic.org/, accessed on 5 March 2022) and for the rest, we used Adobe Photoshop CS6 to make silhouettes of representative species pictures. In this study, we used three secondary calibration points published in previous studies as references (i.e., the most recent common ancestors of *C. l. familiaris* and *V. lagopus*, *V. ferrilata* and *C. l. familiaris* and *B. mutus* and *B. taurus* were calibrated as diverged between 8.8 and 17.95 Ma, 10.13 and 16.86 Ma and 2.85 and 5.6 Ma, respectively) [5,30,31].

### 2.3. Gene Family Expansion and Contraction Analysis

The expansion and contraction of homologous gene families were detected by CAFE v5.0.0 based on the results of OrthoFinder v2.4.0, which uses birth and death processes to simulate gene gains and losses during phylogeny [32]. Gene families that significantly show expansion and contraction in high-altitude species were further analyzed by Gene Ontology (http://www.geneontology.org/, accessed on 5 March 2022) and KEGG (Kyoto Encyclopedia of Genes and Genomes) database enrichment to find gene families and functional categories that convergent evolved in the six high-altitude species.

### 2.4. Rapidly Evolving and Positive Selection Genes Analysis

According to the neutral theory of molecular evolution, the ratio of the nonsynonymous substitution rate (Ka) and synonymous substitution rate (Ks) of protein coding genes can be used to identify genes that show signatures of natural selection. We thus calculated average Ka/Ks values and conducted the two-ratio model using Codeml implemented in the PAML package to identify rapidly evolving genes and positively selected genes in the six high-altitude species [28]. Genes with an ω value of he foreground branch >1 and *p* value < 0.05 under the two-ratio model were considered as positively selected genes. As for rapidly evolving genes, the genes with an ω value of the background branch greater than the ω value of the foreground branch <1 and *p* value < 0.05 were identified as rapidly evolving genes. The results were further analyzed by GO and KEGG pathway enrichment.

## 3. Results

### 3.1. Identification of Orthologous Gene Set and Phylogenetic Analysis

After data processing, an average of 8601 single-copy genes were identified for each of the 15 species, ranging from 8282 (*O. curzoniae*) to 8814 (*O. cuniculus*). These single-copy genes accounted for 30.25% of the total number of genes in each species on average, of which *B. mutus* (44.59%) and *P. hodgsonii* (25.33%) had the highest and the lowest single-copy gene proportion, respectively. More information about the gene sets of the 15 species are shown in Appendix A. These single-copy genes were further used to obtain the one-to-one single-copy orthologous genes. After alignment, a total of 6008 single-copy orthologous genes were identified. After gap removal, 6000 single-copy orthologous genes were used for subsequent analysis (Appendix A).

According to the phylogenetic relationship constructed by single-copy orthologous genes, the six high-altitude species and their closest relatives belong to four different orders, including Lagomorpha, Primates, Carnivora and Artiodactyla (Figure 1). Among them, Artiodactyla has the most high-altitude species, including *B. mutus*, *P. hodgsonii* and *O. aries* (high). Although these three high-altitude species belong to the same family and order (Bovidae and Artiodactyla), the phylogenetic relationship showed that the MRCA (most recent common ancestors) of these three species and their closest relatives were not high-altitude species. Thus, they might have independently evolved high-altitude adaptation mechanisms. According to the results of divergence time estimation, among the six high-altitude species, *O. aries* (high) and its low-altitude close relatives *O. aries* (low) had the closest relationship, with a divergence time of 3.99 Ma (95% CI: 6.30–2.30), while *O. curzoniae* and its low-altitude close relatives *O. cuniculus* had the longest divergence time of 45.15 Ma (95% CI: 65.5–24.9). In general, the results of phylogenetic relationships showed that these six high-altitude species belong to the high-altitude adaptation mechanism of independent evolution, which meets the requirements of convergent evolution research.

### 3.2. Gene Family Expansion and Contraction Analysis

According to the analysis results of gene family expansion and contraction, a total of 17,153 gene families were identified by MRCA (most recent common ancestors) in 15 species for subsequent analysis. Interestingly, except for the *O. curzoniae*, the remaining five high-altitude species had more gene family changes than their closest relatives (Figure 2). It was worth noting that the expanded gene families of the remaining five high-altitude species, except *O. curzoniae*, were not significantly more than their closest relatives, or even less (i.e., 577 of *R. bieti* versus 685 of *T. francoisi*). Compared with low-altitude relatives, the gene family changes of these five high-altitude species had a significantly higher proportion of contracted gene families. At the same time, the number of contracted gene families of high-altitude species was also much larger than that of low-altitude relatives (i.e., 1133 of *R. bieti* versus 600 of *T. francoisi*, 1610 of *P. hodgsonii* versus 708 of *C. hircus*, 2208 of *O. aries* (high) versus 357 of *O. aries* (low), 1991 of *V. ferrilata* versus 482 of *V. lagopus*, and 976 of *B. mutus* versus 409 of *B. taurus*). 

In order to explore the convergence pattern of gene family changes in high-altitude species, gene families that significantly (*p* < 0.05) expanded or contracted together in more than half of the high-altitude species were identified. The results showed that in these six high-altitude species, the convergence pattern of the significant contracted gene families was much larger than the significant expanded gene families (Figure 3). As shown in Figure 3a, 38 significantly expanded gene families were identified to exist in more than 3 high-altitude species, and no significantly expanded gene families were found in all 6 high-altitude species. Furthermore, only 1 of these significantly expanded gene families existed in 5 high-altitude species (OG0000128), and 5 existed in four high-altitude species (OG0000068, OG0000304, OG0000398, OG0001349 and OG0001430). In contrast, 112 significantly contracted gene families were identified to exist in more than 4 high-altitude species (Figure 3b,c). Among the 112 significantly contracted gene families, 2 gene families were identified in 6 species, and 25 gene families were identified in 5 high-altitude species. In addition, among the 450 significantly expanded gene families, the number of species-specific gene families was much higher than that existing in more than 2 species (71 of *P. hodgsonii*, 66 of *O. curzoniae*, 49 of *O. aries*, 36 of *V. ferrilata*, 32 of *R. bieti* and 29 of *B. mutus*, respectively) (Appendix A). As for significantly contracted gene families, compared with the number of gene families existing in more than 2 species, the species-specific gene families did not show a significantly higher trend. Only the number of species-specific gene families of one species was significantly higher than other types of gene families (82 of *P. hodgsonii*) (Appendix A).

To further explore the functional convergence pattern of the significantly changed gene families of these 6 high-altitude species, we performed GO enrichment analysis on the significantly expanded and contracted gene families of each species. Enriched GO terms that existed in at least 5 species were retained for subsequent analysis. As shown in Figure 4a, a total of 29 enriched GO terms existed in at least 5 high-altitude species, of which 16 were enriched in all high-altitude species. In addition, the number of significantly expanded gene families in the same enriched GO term of each species also varied greatly (Appendix A). In contrast to the results of gene families, the gene families that significantly contracted in the 6 high-altitude species did not enrich significantly more GO terms than the enrichment results of the significantly expanded gene families. A total of 37 enriched GO terms existed in at least 5 high-altitude species, of which 11 were enriched in all high-altitude species (Figure 4b and Appendix A).

### 3.3. Rapidly Evolving and Positive Selection Genes

To better understand the effect of high-altitude environment on the nucleic acid sequence changes of genes of different species, the identification of rapidly evolving gens (REGs) and positive selection genes (PSGs) was conducted under the two-ratio model in PAML, and a total of 1713 REGs were identified in all 6 high-altitude species. After KEGG and GO enrichment analysis of these REGs, 40 significantly over-represented pathways and 43 significantly enriched GO terms were obtained (Figure 5a,b). Notably, ultraviolet (UV) radiation response, angiogenesis-related and hypoxia-related REGs showed significant pathway expansion, including the melanoma (ko05218, 19 genes, *p* = 1.69 × 10^−^^6^), MAPK signaling pathway (ko04010, 48 genes, *p* = 2.18 × 10^−^^6^), calcium signaling pathway (ko04020, 38 genes, *p* = 5.41 × 10^−^^5^), EGFR tyrosine kinase inhibitor resistance (ko01521, 15 genes, *p* = 0.000886), regulation of actin cytoskeleton (ko04810, 32 genes, *p* = 0.000986), ubiquitin-mediated proteolysis (ko04120, 24 genes, *p* = 0.001141), Ras signaling pathway (ko04014, 34 genes, *p* = 0.001142), mTOR signaling pathway (ko04150, 24 genes, *p* = 0.002045), PI3K-Akt signaling pathway (ko04151, 44 genes, *p* = 0.002641), renal cell carcinoma (ko05211, 13 genes, *p* = 0.004766) and AMPK signaling pathway (ko04152, 18 genes, *p* = 0.008991). The 43 significantly enriched GO terms, including G protein-coupled receptor signaling pathway (GO:0007186, 28 genes, *p* = 1.13 × 10^−^^20^), G protein-coupled receptor activity (GO:0004930, 26 genes, *p* = 2.69 × 10^−^^19^), potassium ion transport (GO:0006813, 22 genes, *p* = 4.15 × 10^−^^6^), protein phosphorylation (GO:0006468, 71 genes, *p* = 1.18 × 10^−^^5^), voltage-gated potassium channel activity (GO:0005249, 14 genes, *p* = 0.000153), zinc ion binding (GO:0008270, 59 genes, *p* = 0.000341), protein tyrosine kinase activity (GO:0004713, 17 genes, *p* = 0.000551), voltage-gated potassium channel complex (GO:0008076, nine genes, *p* = 0.000814), intracellular signal transduction (GO:0035556, 20 genes, *p* = 0.004235) and ion transport (GO:0006811, 27 genes, *p* = 0.018482) were also related to angiogenesis and hypoxia response and might reflect the convergence adaptation mechanism of these six high-altitude species to the low-oxygen, high UV-radiation and extreme temperature conditions.

The number of PSGs was significantly less compared with REGs. Only seven PSGs were identified in all six high-altitude species. Four of these seven genes might reflect convergent coping mechanisms of high-altitude species when exposed to the stress of high-altitude environments, including CXCL10 gene, ERBIN gene and SCAMP1 gene, which promote angiogenesis under hypoxic conditions, as well as FYCO1 gene, which is involved in UV-radiation damage response (Table 1).

## 4. Discussion

Independent evolutionary species often evolve unique adaptive mechanisms when facing the same selection pressure; however, the previous studies on the mechanism of convergence adaptation at high-altitude were mainly limited to two species or species with a close genetic relationship. Studies covering a wider range of species could help us to understand the most central convergent genetic changes in mammals when facing the selection pressure of a high-altitude environment (i.e., low oxygen content, high UV-radiation, and extreme temperature). In this study, we first expanded the scope of research on the mechanism of convergence adaptation at high altitude to six independently evolved plateau mammals belonging to four orders.

### 4.1. The Convergent Adaption Pattern of Gene Family Expansion and Contraction

During genome evolution, events of gene duplication and gene loss can lead to the contraction and expansion of gene families [33]. Genome changes could be linked to evolutionary processes that cause environmental niche adaptation [34]. By increasing the amounts of protein synthesized or promoting the evolutionary novelty of one of the duplicated genes through sub-functionalization or neo-functionalization, gene duplication might be beneficial [35]. Previous studies on the convergent evolution of different species also indicated that when species face selection pressure such as environment or eating habits, the expansion and contraction of gene families are mainly the result of direction changes [33,36]. In this study, the result of gene family expansion and contraction analysis showed that five high-altitude species have more gene family changes than their closest relatives except for *O. curzoniae*, indicating that the strong selection pressure brought by the environment of the Qinghai-Tibet Plateau makes species have more gene family changes than their relatives to adapt to the plateau environment (Figure 2). Besides, according to the hypothesis that “less is more”, non-functionalization appears more frequently in evolutionary adaptive response [37]. This might have special relevance when populations are faced with changes in selection pressure patterns due to rapid changes of environmental conditions, which also explains that most high-altitude species have more contracted gene families than their low-altitude relatives. As to the contrary results obtained for the order Lagomorpha, this might be due to the distant genetic relationship between *O. curzoniae* and *O. cuniculus*. Ochotonidae and Leporidae have experienced multiple species evolution and differentiation since the common ancestor differentiation, which could also lead to more complex genetic changes [38]. Therefore, the expansion and contraction of the O. cuniculus gene family might have a more complex cause, which also needs to be confirmed by further studies. 

Although a large number of significantly expanded and contracted gene families have been identified in each high-altitude species, there were few gene families with convergence changes, especially those with significant expansion (Figure 3a). Previous studies have shown that mutation is more likely to lead to loss of function rather than acquisition of function, and gene loss makes an important contribution to adaptive evolution, especially rapid response to environmental changes [39]. Another genomics study on yeast also confirmed that gene loss is the main adaptive mechanism to adapt to the growth-restricting environments (e.g., limited amounts of sugar) [40]. In addition, the experiments of bacterial population selection under different conditions showed that the adaptive function loss mutation of enzymes and regulatory functions played an important role in the adaptation of bacterial populations to new environments [41]. In this study, the number of convergent contracted gene families of six high-altitude species was significantly higher than that of convergent expanded gene families, which also indicates that convergent contracted gene families are the main adaptive strategies for high-altitude species to cope with the harsh plateau environment (Figure 3b,c).

Interestingly, there was a large overlap between the GO terms enriched by the significantly expanded and contracted gene families, indicating that the gene families involved in these GO terms have changed greatly (Figure 4a,b). It is worth noting that the gene family related to high-altitude adaptation had both significantly expanded and contracted, such as olfactory receptor activity (GO:0004984) related to hypoxia response [42], the glutathione metabolic process (GO:0006749) [43], ATP hydrolysis activity (GO:0016887) related to energy metabolism and thermogenesis [44] and G protein-coupled receptor activity (GO:0004930 and GO:0007186) related to angiogenesis [45]. In addition, the ion binding gene families (GO:0005509: calcium ion binding and GO:0008270: zinc ion binding) related to hypoxia response has significantly expanded without significant contraction, indicating that the high-altitude species are targeted to cope with hypoxia stress through the expansion of gene families related to calcium and zinc binding [46,47].

### 4.2. The Convergent Adaption Pattern of REGs and PSGs

Previous studies proved that rapid adaptive evolution plays a great driving role for species to adapt to new environments [35]. Understanding what kinds of genes rapidly convergently evolve when independently evolved high-altitude species face the same high-altitude selection pressure can provide valuable insights into mammalian convergence high-altitude adaptation mechanisms. In this study, a considerable number of REGs had been confirmed to be related to the high-altitude adaptation mechanism of species (Figure 5a,b). For example, genes associated with melanoma have been shown to be associated with excessive ultraviolet radiation at high altitudes [48]. Previous studies confirmed that genes involved in the MAPK signaling pathway, calcium signaling pathway, ubiquitin-mediated proteolysis and regulation of actin cytoskeleton are closely related to hypoxia response [49,50,51,52]. The genes involved in EGFR tyrosine kinase inhibitor resistance, the Ras signaling pathway, mTOR signaling pathway, PI3K-Akt signaling pathway and renal cell carcinoma are related to angiogenesis, which has also been confirmed as an important way to cope with hypoxia pressure [53,54,55]. In addition, the gene contained in the AMPK signaling pathway has been proved to be related to energy metabolism, which might be a coping mechanism of species to extreme temperature in a high altitude environment [56]. GO enrichment analysis of REGs yielded similar results, and the enrichment of GO terms for REGs related to G protein-coupled receptors and protein phosphorylation were strongly associated with angiogenesis and energy metabolism [45,57]. The enrichment of GO terms associated with trace elements might be related to the species response to low oxygen pressure, such as potassium and zinc. A previous study has shown that exposure to hypoxia might cause decreased activity of the voltage-gated potassium channel, leading to hypoxia pulmonary vasoconstriction [58]. Another study also confirmed that plasma zinc contents significantly reduced upon exposure to high altitude [46]. In addition to this, the enrichment of REGs related to protein tyrosine kinases and cell signaling transduction might also reflect a convergent coping mechanism of species in the face of hypoxic stress [59,60]. In general, REGs associated with hypoxia stress response, UV-radiation damage repair and energy metabolism might reflect convergent mechanisms underlying the adaptations of six high-altitude species to high-altitude environments.

As for PSGs, a total of seven genes were identified to show a convergent positive selection signature in all high-altitude species. Among these seven genes, three genes might be related to the response of species to hypoxic pressure (Table 1). A previous study showed that Erbb2 interacting protein (*ERBIN*) gene enhanced the expression of Hif-1α protein and the activation of Hif-1α pathway by promoting angiogenesis [61]. Another study indicated that hypoxia/ischemia can significantly increase the production of C-X-C motif chemokine (*CXCL10*) gene in myocardial microvascular endothelial cells [62]. In addition, the down-regulation of the secretory carrier membrane protein 1 (*SCAMP1*) gene inhibited vascular endothelial growth factor (*VEGF*) levels in the conditioned medium of *SCAMP1* siRNA transfected cells [63]. In addition to genes related to hypoxia stress response, we found for the first time that FYVE and coiled-coil domain-containing protein 1 (*FYCO1*) gene possibly related to UV-radiation damage repair showed a convergent positive selection signature in high-altitude species. The formation of cataract is closely related to long-term exposure to UV-radiation. Previous studies have shown that the incidence of cataract in high-altitude residents is much higher than that in plain residents, and the *FYCO1* gene plays a key role in preventing the formation of cataract. Therefore, the convergent positive selection signal of the *FYCO1* gene in high-altitude species might be one of the important mechanisms for species to cope with long-term excessive UV-radiation [64,65]. 

## 5. Conclusions

In this study, we conducted a comparative genomics approach to explore the convergent evolution of high-altitude adaptation mechanisms in six independently evolved mammals belonging to four orders. The results showed that the expanded gene families of these six species did not show significant convergence, while the contracted gene families showed much greater convergence than the expanded gene families, indicating that gene loss rather than gain might be an important adaptation mechanism for species in the face of extreme environments. Nonetheless, enrichment analysis indicated a convergence in the functions of significantly changed gene families across the six species. Besides, the 1713 rapidly evolving genes identified were mainly related to hypoxia stress response, including the activation of pathways and angiogenesis to increase oxygen delivery under hypoxic conditions. These results suggest that oxygen content might be one of the important reasons for the rapid evolution of genes in high-altitude species. In addition, pathways related to UV-radiation damage and energy metabolism might also reflect a coping mechanism of species in the face of excess UV-radiation and extreme temperatures in the plateau environment. Similar results were obtained for the analysis of positively selected genes. Among the four genes that might be related to high-altitude adaptation mechanisms, three positively selected genes are related to hypoxia stress response, and one positively selected gene was related to preventing cataract formation, which is closely related to UV-radiation at high altitude. These results reflected a convergent evolutionary adaptation mechanism of six independently evolving species in the face of the selective pressure of the plateau environment. In general, our study reveals that hypoxia response and UV-radiation might play an important role in the adaptation of independently evolved species to a high-altitude environment, providing a basic perspective for further exploring the high-altitude adaptation mechanism of different related species in the future.

## Figures and Tables

**Figure 1 animals-12-03572-f001:**
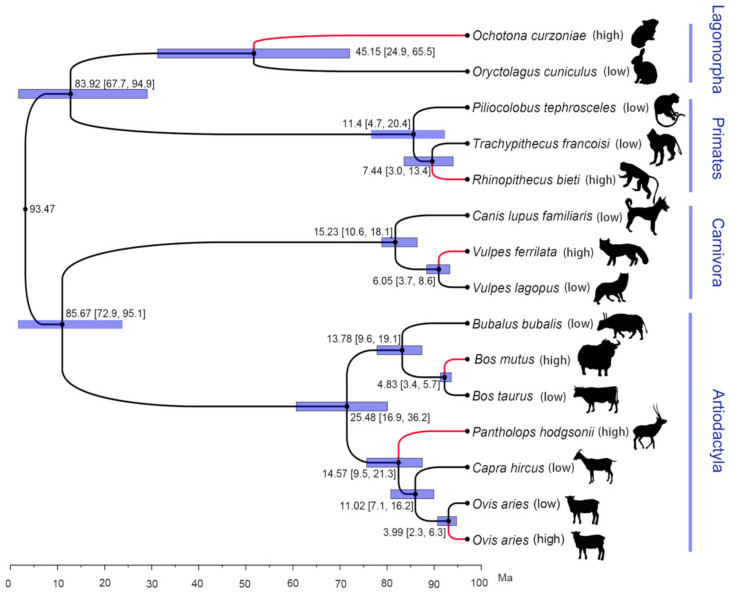
Phylogenetic relationship and divergence time estimation of six high-altitude species and their relatives based on one-to-one homologous genes. The red branches represent species from high altitudes.

**Figure 2 animals-12-03572-f002:**
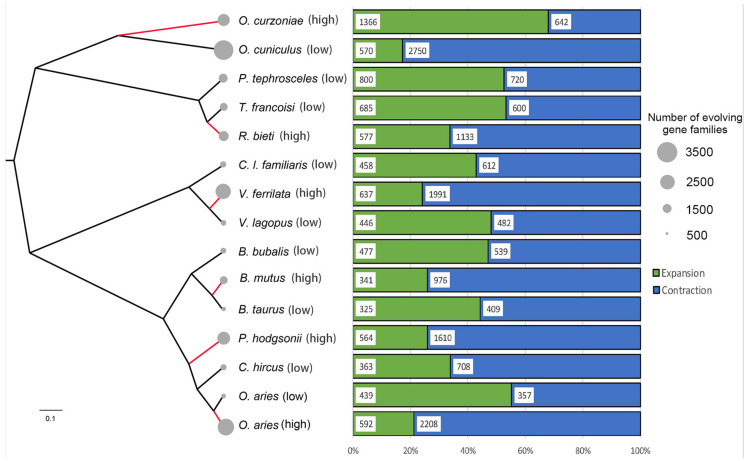
Gene family expansion and contraction in six high-altitude species and their relatives. The different areas of the gray circles to the left of each species name represent the number of gene families changed, where the larger the area, the greater the number of gene families changed. The green and blue rectangles represent the proportion of expanded and contracted gene families in each species to the total number of changed gene families in that species.

**Figure 3 animals-12-03572-f003:**
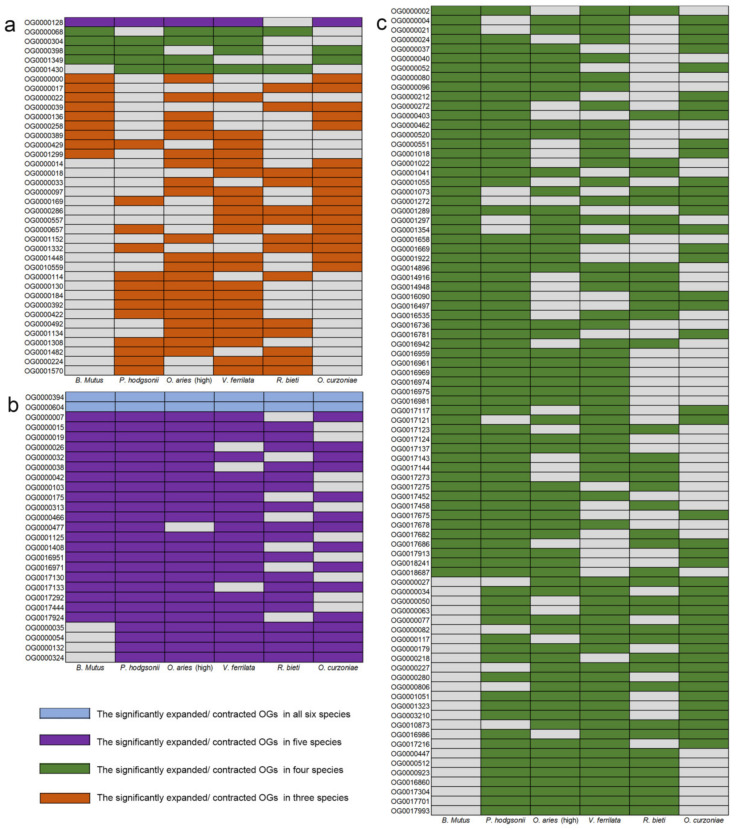
Gene families with significant expansions and contractions in at least three of the six high-altitude species. The gray squares indicate that there is no significant expansion or contraction of this gene family in this species. (**a**) Statistics of gene families with significant convergent expansions in at least three high-altitude species. (**b**) Statistics of gene families with significant convergent contractions in at least five high-altitude species. (**c**) Statistics of gene families with significant convergent contractions in four high-altitude species.

**Figure 4 animals-12-03572-f004:**
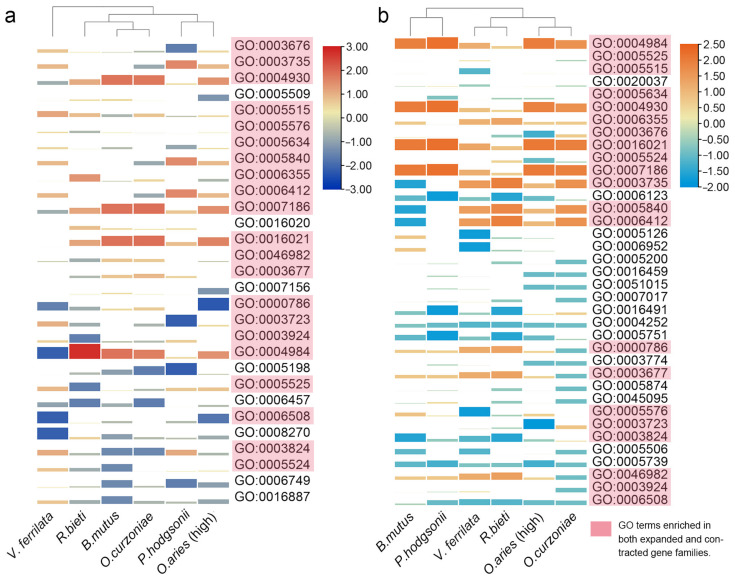
GO terms enriched in at least five high-altitude species were analyzed by clustering heat map. The color and area of each GO term for each species were log transformed from the number of gene families enriched in this GO term for each species. (**a**) Result of GO enrichment for significantly expanded gene families in at least five species. (**b**) Result of GO enrichment for significantly contracted gene families in at least five species.

**Figure 5 animals-12-03572-f005:**
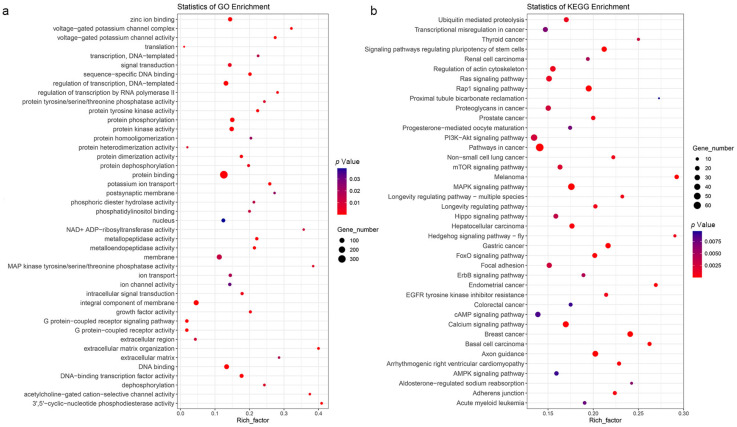
GO term and KEGG pathway enrichment analysis of rapidly evolving genes. The size of the circle in each GO term and pathway of each species represents the number of genes. The color of each circle represents the *p*-value of this term or pathway, and the redder the color, the smaller the *p*-value. (**a**) GO term enrichment analysis of rapidly evolving genes. (**b**) KEGG pathway enrichment analysis of rapidly evolving genes.

**Table 1 animals-12-03572-t001:** Basic information on four genes associated with high-altitude adaptation mechanisms that exhibit convergent positive selection features.

Abbreviation	Full Name	Function
*ERBIN*	Erbb2 interacting protein	Enhanced the expression of Hif-1α protein and promoted angiogenesis
*CXCL10*	C-X-C motif chemokine	Hypoxia response related gene
*SCAMP1*	Secretory carrier membrane protein 1	Vascular endothelial growth factor related gene
*FYCO1*	FYVE and coiled-coil domain-containing protein 1	UV-radiation damage repair related gene

## Data Availability

The high-quality genomes of the 15 species used in this study were obtained from the NCBI database (https://www.ncbi.nlm.nih.gov/, accessed on 5 March 2022), and the accession numbers of the genomes are listed in Appendix A.

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
