# Peer review of "Convergent Genomic Signatures of High-Altitude Adaptation among Six Independently Evolved Mammals"

_animals, 2022, doi:10.3390/ani12243572_

Round 1

Reviewer 1 Report

I really liked the manuscript ant its content. It is also well written and interesting.

I have only a few minor suggestions. First, in line 61 (Introduction) please check English: plural or singular: "two genes play important role" rather than "plays an important ..." On opther parts, English seems to be very good.

 Intorduction is interesting, but this imprtant chapter mostly consists of listing recent findngs and thus after reading it I found it wordy. To Improve Introduction I strongly recommend:

- shorteing it a little bit

 and (most importantly)

- rephase/add last part of Introduction by providing some hypotheses, which are badly needed in this good manuscript.

 For example, Authors mentioned and discussed excelled hypotheses in Discussion: page 11, lines 350-353. By adding hypotheses your paper will be interesting to much broader audience and will be much more cited, as it indeed is worth it!

For Fig 1 and 2 I suggest to make "high" species indicated for clarity and easier reading. Ovies aries high and low are cealry indicated, other "high altitute" species could be shown  on Fig 1 as for example their "violet" icons or, after a Latin name with "H" or "L".

Lood luck with the paper!

Author Response

Thank you very much for your valuable suggestions on our research. The following are our modifications and replies according to the suggestions:

Point 1: Intorduction is interesting, but this imprtant chapter mostly consists of listing recent findngs and thus after reading it I found it wordy. To Improve Introduction I strongly recommend:

- shorteing it a little bit and (most importantly)

- rephase/add last part of Introduction by providing some hypotheses, which are badly needed in this good manuscript.

Response 1: Thank you for your advice.

First, we deleted the overly lengthy literature review in the first paragraph in Intoduction and replaced it with “In addition, a considerable number of studies have confirmed that hypoxia adaptation is the key for species to adapt to the plateau environment, and independently evolved species might have different genetic alterations and mechanistic preferences in response to the same selection pressures [5-8]”.

Then, we then removed the last sentence of the last paragraph in Intoduction and added some hypotheses as requested. The last paragraph after the modification is” Among the six mammals, only the Tibetan sheep has artificially changed its living environment to adapt to the plateau, the yak was domesticated from the plateau wild species, while the rest of the wild species are naturally adapted to the high-altitude environment [3-5, 22-24]. Furthermore, these six animals have great differences in size, feeding habits, phylogenetic relationships and life history. On the one hand, previous studies have shown that in order to cope with the intense selection pressures brought about by the Tibetan Plateau (such as low oxygen content, extreme temperature changes and high ultraviolet radiation), domesticated species and wild plateau species from different phylogenetic backgrounds have unique or identical high altitude adaptation mechanisms [1-8, 18, 19]. On the other hand, the degree of phylogenetic relationships and the number of species selected also play important role in the study of convergence mechanism of species [11, 20]. In other words, the more distantly related the species, the more likely it is that the convergent phenotype or function is the result of different molecular mechanisms, rather than the result of the same gene or amino acid substitution. Therefore, this study provides a new perspective on the mechanisms of high-altitude adaptation by exploring different mammals in the context of large-scale phylogeny, and also expands the understanding of the impact of high-altitude on the genetic changes in mammals”.

Point 2: For Fig 1 and 2 I suggest to make "high" species indicated for clarity and easier reading. Ovies aries high and low are cealry indicated, other "high altitute" species could be shown on Fig 1 as for example their "violet" icons or, after a Latin name with "H" or "L".

Response 2: Thank you for your advice. We have re-labeled the plateau species in Fig. 1 and Fig. 2 to make them easier to distinguish. Please see the attachment for the changed Figures.

Reviewer 2 Report

The authors conducted their research with a focus on the convergent evolution of high-altitude mammals. Unlike other studies so far, this study is meaningful because it selected various species for a more in-depth understanding.

Some comments about the authors' research are presented below.

1) Insufficient information about the mammals the authors selected for the study.

- Is it a perfect wild species from ancient times to the present day? Or is it a species that has been influenced by humans?

-Similarities or differences in the shape, structure, and usage of the six types of body

-Similarities or differences in the lifestyles of the six species

- Similarities or differences in the gene sequences being compared

2) Evidence of convergent or divergent evolution that can be inferred from 1)

3) Definition of convergent evolution: Why is the proposed content convergent evolution? Please give detailed explanation.

-If six species live in the same place, is it convergent evolution?

-If 6 species have the same genes, is it convergent evolution?

-From what difference does the same converge in the six mammal species?

-If there is expansion or contraction of genes, is it necessarily convergent evolution?

Author Response

Response to Reviewer 2 Comments

Thank you very much for your valuable suggestions on our research. The following are our modifications and replies according to the suggestions:

Point 1: 1) Insufficient information about the mammals the authors selected for the study.

- Is it a perfect wild species from ancient times to the present day? Or is it a species that has been influenced by humans?

- Similarities or differences in the shape, structure, and usage of the six types of body

- Similarities or differences in the lifestyles of the six species

- Similarities or differences in the gene sequences being compared

Response 1: Thank you for your advice. We rewrote the last paragraph of the Introduction to add a description of the six selected species and cite some studies similar to this study. The purpose of our study is to find the same high-altitude adaptation mechanism in different species by comparing and analyzing species in the context of large-scale phylogenetic relationships. The difference results unrelated to high-altitude adaptation are not considered.

The rewritten content of the last paragraph is "In this study, we selected six representative mammals at the level of four orders, including Lagomorpha, Primates, Carnivora, and Artiodactyla, in order to compare these six mammals with their low-altitude relatives through the method of comparative genomics, and explored out their convergent evolution at the molecular or functional level when facing the selection pressure of high-altitude. Among the six mammals, only the Tibetan sheep has artificially changed its living environment to adapt to the plateau, the yak was domesticated from the plateau wild species, while the rest of the wild species are naturally adapted to the high-altitude environment [3-5, 22-24]. Furthermore, these six animals have great differences in size, feeding habits, phylogenetic relationships and life history. On the one hand, previous studies have shown that in order to cope with the intense selection pressures brought about by the Tibetan Plateau (such as low oxygen content, extreme temperature changes and high ultraviolet radiation), domesticated species and wild plateau species from different phylogenetic backgrounds have unique or identical high altitude adaptation mechanisms [1-8, 18, 19]. On the other hand, the degree of phylogenetic relationships and the number of species selected also play important role in the study of convergence mechanism of species [11, 20]. In other words, the more distantly related the species, the more likely it is that the convergent phenotype or function is the result of different molecular mechanisms, rather than the result of the same gene or amino acid substitution. Therefore, this study provides a new perspective on the mechanisms of high-altitude adaptation by exploring different mammals in the context of large-scale phylogeny, and also expands the understanding of the impact of high-altitude on the genetic changes in mammals.".

Point 2: 2) Evidence of convergent or divergent evolution that can be inferred from 1)

Response 2: Thank you for your advice. Our response is consistent with point 1

Point 3: 3) Definition of convergent evolution: Why is the proposed content convergent evolution? Please give detailed explanation.

- If six species live in the same place, is it convergent evolution?

- If 6 species have the same genes, is it convergent evolution?

- From what difference does the same converge in the six mammal species?

- If there is expansion or contraction of genes, is it necessarily convergent evolution?

Response 2: Thank you for your advice. The following is our reply and modification:

Definition: Convergent evolution refers to the evolution of different lineages that are similar in structure or function, and this evolution has nothing to do with the existence of a common ancestor between species. Simply put, the fact that structure or function are similar does not reflect homology [1]. Strictly speaking, convergence at the molecular level refers to the substitution of the same alleles with independent origins in different lineages, that is to say, convergent evolution refers to the independent conversion of different ancestral amino acids to the same derived amino acids at specific sites [2]. In broad terms, however, convergence may exist at the phenotypic or functional level, in which lineages share dominant traits, but the molecular mechanisms that achieve this convergence trait are different between species [3].

- If six species live in the same place, is it convergent evolution?

Convergent evolution occurs when species occupy similar ecological niches and adapt in similar ways to similar selection pressures [4, 5]. Therefore, convergent evolution is possible only when species are subjected to the same selective pressures.

- If 6 species have the same genes, is it convergent evolution?

In this study, we calculated average Ka/Ks values and conducted the two-ratio model using Codeml implemented in the PAML package to identify rapidly evolving genes and positively selected genes in the six high-altitude species. Compared with the branch-site model, the filtration standard of the two-ratio model is more stringent. Another difference is that the positive selection gene screened by the two-ratio model does not mean that the gene has the same amino acid substitution in all the six species. Therefore, it is not necessary to discuss whether these genes are convergent evolution or parallel evolution in a strict sense. In addition, previous studies have suggested that parallel substitution of amino acid sites leads to the generation of convergent phenotypes [6]. In a broad sense, we think that the functions of these genes evolved convergent across the six species in order to adapt to the selective pressures of high-altitude.

- From what difference does the same converge in the six mammal species?

In the section on expansion and contraction of gene families, we discuss gene families that have changed significantly in all six species, and it turns out that only a small number of gene families have expanded or contracted in all six species. On the other hand, the same gene family changes do not mean that the gene family has the same gene loss or gain in all six species. However, the results of functional enrichment of the significantly altered gene families in each species showed that these gene families were enriched in the same GO term, and the previous results suggested that a part of the GO term may be related to the plateau adaptation mechanism of the species.

- If there is expansion or contraction of genes, is it necessarily convergent evolution?

There are many reasons for the expansion or contraction of gene families in species, most of which have nothing to do with convergent evolution. In this study, we only discuss the enrichment results of significant gene family changes, looking for parts that may be related to the adaptation of species to high altitude.

The references cited in the response:

[1] Gabora, L. Convergent evolution. Brenner's Encyclopedia of Genetics. 2013, 2, 178-180.

[2] Zhang, J., Kumar, S. Detection of convergent and parallel evolution at the amino acid sequence level. Mol. Biol. Evol. 1997, 14, 527-536.

[3] Wu, D.D., Yang, C.P., Wang, M.S., Dong, K.Z., Yan, D.W., Hao, Z.Q., Fan, S.Q., Chu, S.Z., Shen, Q.S., Jiang, L.P., Li, Y., Zeng, L., Liu, H.Q., Xie, H.B., Ma, Y.F., Kong, X.Y., Yang, S.L., Dong, X.X., Esmailizadeh, A., Irwin, D.M., Xiao, X., Li, M., Dong, Y., Wang, W., Shi, P., Li, H.P., Ma, Y.H., Gou, X., Chen, Y.B., Zhang, Y.P. Convergent genomic signatures of high-altitude adaptation among domestic mammals. Natl. Sci. Rev. 2020, 7, 952-963.

[4] Houliez, E., Lefebvre, S., Dessier, A., Huret, M., Marquis, E., Bréret, M., Dupuy, C. Spatio-temporal drivers of microphy-toplankton community in the Bay of Biscay: Do species ecological niches matter? Prog. Oceanogr. 2021, 194, 102558.

[5] Stern, D.L. The genetic causes of convergent evolution. Nat. Rev. Genet. 2013, 14, 751-764.

[6] Xu, D., Yang, C., Shen, Q., Pan, S., Liu, Z., Zhang, T., Zhou, X., Lei, M., Chen, P., Yang, H., Zhang, T., Guo, Y., Zhan, X., Chen, Y., Shi, P. A single mutation underlying phenotypic convergence for hypoxia adaptation on the Qinghai-Tibetan Plateau. Cell Res. 2021, 31, 1032-1035.

We have rewritten the content of the second paragraph and the new content is “Convergent evolution refers to the evolution of different lineages that are similar in structure or function, and this evolution has nothing to do with the existence of a common ancestor between species. Simply put, the fact that structure or function are similar does not reflect homology [9]. Strictly speaking, convergence at the molecular level refers to the substitution of the same alleles with independent origins in different lineages, that is to say, convergent evolution refers to the independent conversion of different ancestral amino acids to the same derived amino acids at specific sites [10]. In broad terms, however, convergence may exist at the phenotypic or functional level, in which lineages share dominant traits, but the molecular mechanisms that achieve this convergence trait are different between species [11]. In addition to the individual adaptation mechanisms that species in different phylogenetic contexts develop when faced with the same selection pressure, the same selection pressure also leads to convergent evolution, including phenotypic convergence as well as genetic and functional convergence at the molecular level. Convergent evolution occurs when species occupy similar ecological niches and adapt in similar ways to similar selection pressures [12, 13].”
